# Advances in Shear Stress Stimulation of Stem Cells: A Review of the Last Three Decades

**DOI:** 10.3390/biomedicines12091963

**Published:** 2024-08-29

**Authors:** Qiyuan Lin, Zhen Yang, Hao Xu, Yudi Niu, Qingchen Meng, Dan Xing

**Affiliations:** 1Arthritis Clinical and Research Center, Peking University People’s Hospital, No.11 Xizhimen South Street, Beijing 100044, China; qiyuanlin_pku@163.com (Q.L.); zhenyang_pku@163.com (Z.Y.); 2110122442@bjmu.edu.cn (H.X.); mengqingchen_0406@163.com (Q.M.); 2Arthritis Institute, Peking University, Beijing 100044, China; 3Department of Biomedical Engineering, School of Medicine, Tsinghua University, Beijing 100084, China; nyd20@mails.tsinghua.edu.cn

**Keywords:** stem cell, shear stress, bibliometric, visualization

## Abstract

Stem cells are widely used in scientific research because of their ability to self-renew and differentiate into a variety of specialized cell types needed for body functions. However, the self-renewal and differentiation of stem cells are regulated by various stimuli, with mechanical stimulation being particularly notable due to its ability to mimic the physical environment in the body. This study systematically collected 2638 research papers published between 1994 and 2024, employing tools such as VOSviewer, CiteSpace, and GraphPad Prism to uncover research hotspots, publication trends, and collaboration networks. The results indicate a yearly increase in global research on the shear stress stimulation of stem cells, with significant contributions from the United States and China in terms of research investment and output. Future research directions include a deeper understanding of the mechanisms underlying mechanical stimulation’s effects on stem cell differentiation, the development of new materials and scaffold designs to better replicate the natural cellular environment, and advancements in regenerative medicine. Despite considerable progress, challenges remain in translating basic research findings into clinical applications.

## 1. Introduction

For multicellular organisms, stem cells are undifferentiated or partially differentiated cells with a perpetual potential for self-renewal as the earliest cell lineage during development, and they can develop into many different types of cells in the body during early life and growth [1]. With their remarkable differentiation ability, stem cells are of great clinical potential for medical and especially cell therapy applications. Cell therapy is the transplantation of human cells to repair or replace diseased and damaged tissues or cells. Cellular therapy represented by stem cells such as hematopoietic stem cells [2] and mesenchymal stem cells (MSCs) [3] has broad development prospects in the treatment of hematological malignancies, neurodegeneration, etc. [4]. However, the applications of stem cells face a challenge to be addressed: how to regulate the self-renewal and differentiation of stem cells. The self-renewal and differentiation of stem cells are regulated by a variety of stimulation, including biochemical and mechanical stimulation. The regulatory role of various bioactive substances in the process of stem cell proliferation and differentiation has been widely studied, but the biochemical mechanisms of action are complex and extremely finely tuned, and even small perturbations in culture can lead to undesirable results. In contrast, an increasing number of studies have shown that mechanical stimulation is likely to control the fate of stem cells in vitro [5] due to their imitation of the physical environment in the body and lower regulatory costs. The mechanical properties and structure of the matrix, cyclic mechanical strain, and fluid shear stress all play an important role in stem cell differentiation. Substrate elasticity has been shown to strongly influence stem cell behavior. For instance, gels with an elasticity of 100 to 500 Pa initiate neuronal cell differentiation for neural stem cells, while stiffer gels between 1000 to 10,000 Pa favor glial cell differentiation [6]. The effect of nanoscale topography on stem cell behavior has also been shown to play different roles in regulating the direction and efficiency of stem cell differentiation in different nanotopography [7]. Microgravity is also a prospective way to tune the differentiation of MSCs. Although microgravity could result in disordered bone metabolism and inhibit osteogenic differentiation by decreasing the activation of TAZ, a transcriptional coactivator with a PDZ binding motif [8], simulated microgravity was reported to promote the differentiation of MSCs toward the endothelium and neurons [9]. In addition, external forces can influence the fate of stem cells by regulating gene expression and thus the fate of stem cells. For example, the expression of specific genes in MSCs (mesenchymal stem cells) changes when subjected to external forces in different directions [10]. In addition, cyclic mechanical strain with 1 Hz frequency at a magnitude of 10 for 24 h facilitated MSC differentiation toward vascular smooth muscle cells, but 0.1 Hz led to differentiation toward osteoblasts [11]. Meanwhile, Huang and Lu found that the amplitude and frequency of fluid shear stress affected the differentiation direction of MSCs [12,13]. In general, many results have been obtained regarding the regulation of stem cell differentiation by biochemical and biomechanical stimulation, but both stimulation modalities have advantages and disadvantages. One possible direction is to study the effect of biochemical combined with biomechanical stimulation on stem cell differentiation, and there are fewer related studies [14]. Second, in addition to the tissue engineering approach, the research results on the regulation of stem cell differentiation by mechanical stimulation may also provide new prospects for the treatment of degenerative diseases such as arthritis [15]. In addition, the matrix property is key to biomechanical stimulation, but the molecular mechanism of the associated mechanotransduction is not clear, which is crucial to the development of regenerative medicine and requires collaborative research among clinicians, biologists and material scientists [16]. Although the effects of mechanical stimulation on stem cells have been extensively investigated, new research directions and applications are still unclear, which inconveniences researchers in the field. Moreover, the study of the effects of mechanical stimulation on stem cell differentiation increasingly relies on interdisciplinary and multiregional collaborations, but there is a gap in research regarding the collaborative relationships of researchers in this field, which hinders the full collaboration of researchers in this field. This study therefore aims to fill this gap by providing a systematic and generalized analysis of publication trends, research frontiers, and collaborative relationships.

In recent years, bibliometric methods have been widely used to mine the information behind large amounts of scientific literature data to assist in research decisions. More importantly, combined with visualization methods (e.g., CiteSpace and VOSviewer), trends of publications, research prefaces, and hotspot predictions in specific research areas can be presented in a vivid and clear manner. There is still a lack of articles that holistically analyze the publication trends of stem cell mechanical stimulation. Accordingly, in this study, a new bibliometric combined with visualization analysis will complement this knowledge gap. This study analyzes the literature on stem cell mechanical stimulation over the last three decades (1994–2024) and performs a visual analysis to characterize and predict research trends in the field.

## 2. Result

We compiled 2638 research entries centered on the topic of shear stress stimulation on stem cells, drawing from the Web of Science Core Collection (WoSCC) database and covering the period from 1994 to 2024. Analytical tools such as VOSviewer, CiteSpace, and GraphPad Prism were employed, and we conducted comprehensive analyses to discern the general characteristics, historical evolution, key literature, and pivotal keywords within this research field. This comprehensive analysis enabled us to identify emerging focal points and current trends in the topic of shear stress stimulation on stem cells.

## 3. Trends of Global Literatures

A total of 2758 papers were collected from the WoSCC database and entered into the next analysis based on the search formula for the time interval 1994 to 2024. Next, 101 non-research or review documents were excluded, leaving 2105 research documents and 552 review documents for further analysis. Then, 19 non-English studies were screened out, and the remaining 2638 studies were analyzed and visualized using GraphPad Prism 8, Origin 8, VOSviewer 1.6.20, and CiteSpace 6.3 software (Figure 1). After the literature screening, we analyzed the trends of publications on stem cell mechanics stimulation over the past three decades, the number of publications by country, and the future publication trends. As shown in Figure 2A, global publications on this topic have risen yearly over the past three decades, from two in 1995 to one hundred ninety-eight in 1994, with the highest number of publications in 2022, when two hundred thirty-five articles on the topic were published. In addition, the relative research interest (RRI) in this field has been rapidly increasing each year since 2006 and has remained at a high level in recent years.

In general, 70 countries/regions contributed to this field, and the main countries are marked in Figure 2B,C (the figure counts the total number of US publications as 100%). The number of publications in the USA (1065, 100%) is far ahead of other countries/regions. The number of publications from second to fifth place is as follows: China (462, 43.4%), Germany (211, 19.8%), England (180, 16.9%), and South Korea (133, 12.5%). As shown in Figure 2D, the number of articles published by the top ten countries/regions has fluctuated over the past three decades, but the number of articles published by the U.S. has slightly decreased over the past two years, while the number of articles published by China in this field has continued to increase. As shown in Figure 2E, to better predict future global trends in publishing, this study used a logistic regression model to plot the temporal curves of posting volume, and the correction coefficient R^2^ is 0.9899.

## 4. Quality Analysis of Publications by Countries/Regions

The number of citations and h-index can reflect the quality of research to a certain extent. As shown in Figure 3A, the USA (57,371) has the highest total citations and is substantially ahead of other countries. China came in second place (15,091), followed by Germany (10,003), England (6931), and Canada (6281). The leadership of the USA in this field was also reflected in the h-index (114), with China (62), Germany (54), England (44), and Canada (41) behind it (Figure 3C). In addition, as shown in Figure 3B, the USA (53.9) has the highest average citation, and Canada (47.9) ranked second, followed by Italy (47.5), Germany (47.4), and England (38.5). In summary, countries with stronger research have an advantage in terms of the number of articles and the quality of articles. This suggests that researchers in this field can focus more on these countries when following relevant research progress and academic conferences.

## 5. Collaboration Analysis of Countries/Regions and Institutions

This study also analyzed the cooperative relations of countries around the world in this field. From the size of the nodes representing countries in Figure 4A,C, it can be seen that the United States (367,377) has the most international cooperation, while China (183,754), England (106,954), Germany (99,022), and South Korea (63,352) also have more international cooperation. Among them, the intensity and results of the international cooperation of the United States far exceed those of other countries. In addition, the color of the nodes represents the distribution of cooperation time. From the point of view of partners, the partners of England in international cooperation are mainly Germany, France, and Portugal.

Regarding publication volume, Table 1 identifies the top 10 institutions in terms of contributions, with the University of California system at the forefront, evaluated by article count. An analysis of international collaborations between institutions placed Harvard University at the top (Figure 4D). As depicted in Figure 4B, VOSviewer was used to estimate research impact based on citations for the institutions, with Harvard University having the highest impact (link strength 41,888), followed by Massachusetts Institute of Technology (link strength 34,383) and Stanford University (link strength 28,393); notably, Columbia University demonstrated a strong research impact and had the highest research output, leading in research impact.

## 6. Analysis of Journals and Research Areas

Table 2 shows the top ten journals with the most articles related to shear stress stimulation on stem cells, mainly as follows: the journal Biomaterials published most of the papers (71), and the journal Acta Biomaterials ranked second with 54 papers, followed by Plos One (51), Scientific Reports (45), and Lab on A Chip (44). Regarding the citation relationship, the dual-map overlay of journals related to shear stress stimulation on stem cells is drawn in Figure 5A, which describes four major citation paths marked in pink and orange. It reflected the concentration of research in chemistry, materials, and physics. Co-citation and bibliographic coupling relationships are shown in Figure 5B,C, where only journals with more than 20 citations were included. The top five journals with the highest total link strength were Biomaterials (822,133), Proc Natl Acad Sci USA (521,478), Nature (394,669), Science (321,160), and Scientific Reports (297,137). Then, we detected the burst of journal citations using CiteSpace, and the top 25 journals with the strongest citation burst are shown in Figure 5D, where Blood, Exp Hematol, Am J Pathol, and J Immunol had the longest burst of citations, 17 years from 1995 to 2011, and the minimum burst duration was only 8 years. In addition, the highest citation intensity was Tissue eng (60.35), followed by J Appl Physiol (49.11), J Biomed Mater Res (46.25), J Biol chem (42.81), and Science (40.09).

Regarding research areas, the most representative research areas in Shear Stress Stimulation on stem cells (Table 3) include Engineering, Materials Science, Cell Biology, Science Technology, Other Topics, and Biochemistry Molecular Biology. The analysis of research areas can reflect the research directions that have received the most attention and achieved the most progress in recent years.

## 7. Analysis of Authors

The top 10 authors with the most publications and citations are shown in Table 4. For the collaboration and co-authorship analysis, only authors with more than 20 citations were considered, and a total of 1042 authors met this condition, using VOSviewer and CiteSpace for the analysis (Figure 6A–C). It is shown in Figure 6A,C that Engler AJ has the most citations (339) and the strongest association (10,240) with other authors. Ingber DE (citations = 196, link strength = 7459), Yamamoto, K (citations = 245, link strength = 5952), Sikavitsas, VI (citations = 148, link strength = 5191), and Wang, H (citations = 183, link strength = 5129) also have stronger associations with other authors. From Figure 6B, we can see that authors with more collaborations have maintained a steady amount of collaboration over time, and in addition, it can be found that many new author collaborations have arisen in this field in recent years, indicating that more researchers are beginning to pay attention to and work on the development of mechanical stimulation for stem cells. Moreover, the top 25 cited authors with the strongest citation burst were detected using CiteSpace. It is shown in Figure 6D that Pittenger MF had the strongest burst of citations (30.94), and Yamamoto, K (29.7) ranked second, followed by Kreke MR (23.02), Zhang Y (22.45), and Lee J (20.36). Pittenger MF had a maximum burst duration from 2004 to 2014, while the minimum duration was 3 years.

## 8. Analysis of Funds and References

A list of the top 10 funding sources related to mechanical stimulation on stem cells is presented in Table 5. As is shown, the National Institutes of Health and United States Department of Health Human Services both supported the most articles about mechanical stimulation on stem cells, with 596 articles. The top three funds are from China and the United States, and they together supported 56.762% of the articles in this field, which may serve as an explanation for the leadership of the United States and China in the number and quality of articles published in this field.

The top five review articles related to mechanical stimulation on stem cells and the top five research articles are listed in Table 6 and Table 7. In addition, the co-citation relationship of the references was analyzed in the present study using VOSviewer, as shown in Figure 7A. The top 25 preferences with the strongest citation burst were identified by CiteSpace and are summarized in Figure 7B.

## 9. Analysis of Keywords and Hotspots

The keyword analysis was conducted by VOSviewer. As shown in Figure 8A,B, the five most frequently used keywords were differentiation (frequency = 604), in vitro (574), expression (266), extracellular-matrix (233), and osteogenic differentiation (225). These high-frequency keywords are likely to be the research hotspots in the field and are more likely to achieve research findings with high citation. Furthermore, in Figure 8B, we can see the changes in research directions in the field of mechanical stimulation on stem cells by the appearance time of keywords. The purple nodes represent research directions that once received attention, the yellow nodes represent emerging research directions in which new research hotspots may occur, and the green nodes that occupy a large area represent current research hotspots that have been around for some time, received attention from researchers, and produced more results. We also detected the top 25 keywords with the strongest citation burst using CiteSpace, as shown in Figure 8D.

The clustering analysis of the keyword network was visualized by CiteSpace, and each node represents a hot keyword. Figure 8C shows that the ten most popular keywords were #0 tissue engineering, #1 mesenchymal stem cells, #2 expression, #3 endothelial progenitor cells, #4 model, and #5 stem cells.

## 10. Discussion

With the increasing understanding of stem cells, the possibility of stem cell therapy for intractable diseases is gaining more attention from researchers [41]. Relatedly, research on shear stress stimulation on stem cells has also started to emerge in the last three decades and has gradually become a hot spot for multidisciplinary research, with a large number of results emerging from basic and preclinical studies [42,43]. Therefore, this study focuses on the field of shear stress stimulation on stem cells, using bibliometric and visualization methods to analyze the literature for a total of three decades from 1994 to 2024, and provides a comprehensive survey of the research progress and trends in this field.

## 11. Development Overview of Shear Stress Stimulation on Stem Cells

This study shows that articles about mechanical shear stress stimulation on stem cells have seen an extremely significant increase over the last three decades, from two in 1995 to one hundred ninety-eight in 2023. In addition, the relative research interest in the field is also in growth overall, except for a slight decline in three years, 2012, 2013, and 2019. In terms of national or regional assessments of research output, 70 countries/regions have published scientific papers in the field of shear stress stimulation on stem cells. Total publication number, total citations, average citations, and h-index are all valid parameters for the bibliometric assessment of a country’s scientific research strength in a specific field. Total publications and total citations are more indicative of research investment and scale by country. The h-index in each country basically corresponds to the two parameters with the USA in the absolute leading position. However, the average citation can reflect the research quality more effectively. By comparison, this study found that the USA is the relative leader in terms of research scale and quality. In contrast, China and England, which perform better in terms of the number of publications, total citations, and h-index, are ranked lower in terms of average citations, indicating that there is a contradiction between the scale and quality of research in this field that deserves further study. Moreover, Canada ranked fifth in the total number of publications and first in terms of average citations, suggesting a higher quality of research, but it may need more investment to scale up research in this field.

As shown in Table 1, the ten research institutions with the most publications are mainly from the USA, China, and Europe. Together, these institutions produced over 20% of the papers in this field. The average citation and h-index of each institution was higher, indicating the strong research capability of these institutions. As shown in Table 5, the ten funds supporting the most research in this area are from the United States (4), Europe (4), China (1), and Japan (1), and together, these funds supported approximately 83% of the research in this area, which is consistent with the previous analysis regarding the quality and quantity of national research.

From the above analysis, it is clear that research on shear stress stimulation on stem cells is in a stage of continuous development and growth. The countries that have made major contributions in this field have high-quality research institutions and sufficient funds to support the related research, so other countries can also increase their funding and foster high-quality research institutions. Further progress is intended to be made in the field of shear stress stimulation on stem cells.

## 12. Quality of Authors, Journals, and Articles

The top ten authors that published the most papers about shear stress stimulation on stem cells over the last three decades are summarized in Table 4. The top authors are mainly from the United States of America, and in addition, the most contributing fund to the field, as mentioned above, is the National Institutes of Health, which implies that there is a huge investment in the field in the United States of America and the emergence of highly productive researchers. However, there are limitations in ranking authors only by the number of publications, which does not fully reflect the influence of authors on shear stress stimulation on stem cells. Therefore, this study analyzed and visualized the co-collaborative and cooperative relationships. As seen from Figure 6A–C, Engler AJ (citations = 339, link strength = 10,240), Ingber DE (citations = 196, link strength = 7459), Yamamoto, K (citations = 245, link strength = 5952), Sikavitsas, VI (citations = 148, link strength = 5191), and Wang, H (citations = 183, link strength = 5129) had the most citations and collaborations among all authors, which means they have a stronger influence in the field. Hence, one way to quickly understand the latest advances and research hotspots in shear stress stimulation on stem cells is to focus more attention on the papers of these influential and productive authors.

This study also provides an in-depth analysis of the journals. Table 2 shows the ten journals with the highest output, which together publish approximately 18% of the articles in the field. Among them, Biomaterials, Acta Biomaterialia, and Plos One led the other journals in the number of articles published on the topic of shear stress stimulation on stem cells, and the three journals had higher total citations, average citations, and h-index than most journals. Therefore, these high-output journals are a reasonable choice for the future publication of high-quality papers. The citation analysis and co-citation analysis of the journals are shown in Figure 5A–C. Among them, the dual-map analysis indicates that most of the studies in this field were concentrated on Physics, Materials, Chemistry, Molecular, Biology, and Genetics. For co-citation analysis, it is shown in Figure 5C that Biomaterials had the strongest co-citation relationship with other journals, which means it played a vital role in research about shear stress stimulation on stem cells.

To analyze the impact of published papers, the top five most influential review articles and research articles are listed in Table 6 and Table 7, respectively. Table 6 shows that the review article with the most citations was “Mechanobiology of YAP and TAZ in physiology and diseases”, and in Table 7, “Direct 3D Printing of Shear-Thinning Hydrogels into Self-Healing Hydrogels” is one of the most cited studies, which indicated that research on tissue engineering and bone regenerative medicine has received great attention. Among the ten highly cited studies in Table 6 and Table 7, basic research focused on biomaterials, tissue engineering, and regenerative medicine was still the main type of research.

It is worth noting that the co-citation analysis of references can effectively reflect which papers have contributed the most to the field. In the present study, the visualization of the co-citation analysis of references was conducted and is shown in Figure 7A. “Matrix elasticity directs stem cell lineage specification” authored by Enlger Aj et al. was the top reference with the strongest co-citation relationship. The top 25 references with the strongest citation burst were mainly related to basic research, including repair, gene expression, tissue engineering, and mechanical induction, demonstrating that these directions were hot topics in this field.

## 13. Research Hotspots and Frontiers

In the present study, co-occurrence analysis, clustering analysis, and citation burst of keywords were conducted to reflect the hotspots and frontiers in the field of shear stress stimulation on stem cells. From Figure 8A, the highest frequency of the keyword “differentiation” indicates its popularity in stem cell studies. As early as 2002, Altman GH et al. demonstrated that mechanical stimulation in vitro in the absence of ligament-selective exogenous growth and differentiation factors can induce the differentiation of bone marrow mesenchymal cells into a chondrocyte lineage [44]. Recently, Zhang M et al. used hydrogels with tunable mechanical properties to modulate cellular spatial sensing, regulate mechanotransduction and promote the osteogenic differentiation of MSCs on soft hydrogels [45]. According to the keywords in the titles and abstracts of the included papers, in the current study, we charted the distribution of the average occurrence time of keywords (Figure 8B), which shows the change in concrete research directions over time.

The above keyword analysis can not only help researchers understand the development of shear stress stimulation on stem cells and recognize the current research hotspots but also predict the main research directions in the future, which are further detailed as follows.

(I)Differentiation:

The keyword “differentiation” emerged as the most frequently occurring term, highlighting its central importance in the research domain of shear stress stimulation on stem cells [46]. Differentiation refers to the process by which stem cells transform into specific, functional cell types under certain conditions [47]. The role of shear stress in influencing stem cell differentiation is a critical area of study, as it provides insights into the mechanotransduction mechanisms that govern cell fate decisions [48].

Shear stress, as a form of mechanical force, impacts stem cell differentiation through various pathways and mechanisms. It can alter the cytoskeleton [49], cell adhesion properties [50], and activate specific signaling pathways, such as Wnt, Notch [51], and MAPK [52]. These pathways are crucial in directing the lineage-specific differentiation of stem cells. For instance, research has shown that applying appropriate shear stress to mesenchymal stem cells (MSCs) can promote their differentiation into osteoblasts [53] or chondrocytes [54], making shear stress a significant factor in tissue engineering and regenerative medicine.

Moreover, the interaction between shear stress and the extracellular matrix (ECM) is also pivotal in stem cell differentiation [55]. The ECM provides biochemical cues that, when combined with mechanical signals from shear stress, further regulate stem cell fate. Shear stress influences the cytoskeleton organization and ECM rigidity, activating mechanotransducers like YAP and TAZ, which translocate to the nucleus and modulate gene expression, thereby controlling cell differentiation and proliferation [56].

(II)Mechanism study:

The regulation of stem cell fate is influenced by a complex combination of biochemical effects and mechanical stimulation in the microenvironment in which they are located. Existing methods for controlling stem cell fate do not fully recapitulate the natural microenvironment and corresponding mechanotransduction signaling pathways, especially in terms of drug delivery, the temporal control of responses, and spatial control [57]. Accordingly, researchers have proposed the use of particles, especially nanoparticles, to achieve these high-precision modulations. These particles can be used as carriers due to their controllable size and porous and dispersible properties, offering the possibility of targeted signaling pathways and temporal-spatial control of the response [58]. Some results have been achieved by researchers in 2D scale simulations. Existing research has focused on the incorporation of particles into cellular scaffolds to simulate the corresponding microenvironment in 3D structures [59]. In the future, research on particles for biosignaling and the simulation of cellular microenvironments will remain a hot topic. In addition, understanding the behavior of particles in complex 3D environments and exploring the mechanisms of particle effects on cell–cell signaling are of increasing interest to researchers.

(III)Endothelial cell:

Endothelial cells refer to simple squamous epithelial cells mainly located in the heart, blood vessels, and lymphatic vessels. Under normal conditions, endothelial cells not only serve as an anticoagulation barrier and selective permeability barrier between the vessel wall and blood but also play a critical role in metabolism and synthesis and are able to regulate the immune response, vascular tone, and angiogenesis of the body [60]. In addition, endothelial cell abnormalities are closely related to a variety of diseases, especially those of the circulatory system, such as atherosclerosis, coronary artery disease, heart valve lesions, peripheral vascular disease, renal failure, tumor growth, and severe viral infectious diseases [61].

The study of stem cell differentiation into endothelial cells has thus received much attention, and the results may be applied both to cellular therapies and to contribute to a deeper understanding of the role of endothelial cells in related diseases. James, D. et al. [62] and Patsch, C. et al. [63] achieved stem cell differentiation into endothelial cells by different biochemical stimuli, while Vander Roest MJ. et al. [64] experimentally verified that cyclic strain can promote stem cell differentiation into endothelial cells by increasing the expression of long-stranded noncoding RNA H19. Future research in this direction may focus on regulating the differentiation of stem cells to endothelial cells through mechanical stimulation, such as the mechanical stimulation of different natures and frequencies and bionic mechanical stimulation, to solve the specific problems of cell therapy applied to endothelial cells.

(IV)Mechanotransduction:

In the context of shear stress stimulation on stem cells, the term mechanotransduction is of paramount importance. Mechanotransduction refers to the process by which cells convert mechanical signals into biochemical signals [65]. This process is particularly significant in stem cell research, as the behavior and fate of stem cells can be regulated by mechanical stimulation. Shear stress, as a vital mechanical force, can influence stem cell proliferation, differentiation, and migration through mechanotransduction [66].

Under shear stress stimulation, stem cells perceive mechanical forces via their surface receptors, such as integrins [67] and ion channels [68]. These receptors initiate a cascade of intracellular events that convert the mechanical stimulus into biochemical signals [67,68]. The initial mechanical signal leads to cytoskeletal reorganization, which involves the restructuring of actin filaments and microtubules, and the formation of focal adhesions [69]. This reorganization is crucial for transmitting mechanical forces from the cell surface to the interior of the cell.

One of the key components in mechanotransduction is the activation of signaling pathways such as the MAPK/ERK pathway, the PI3K/AKT pathway, and the YAP/TAZ pathway. These pathways play significant roles in regulating gene expression, cell survival, and differentiation. For example, the YAP/TAZ pathway, which is sensitive to mechanical cues, can translocate to the nucleus and interact with transcription factors to modulate gene expression [70]. This modulation influences stem cell fate decisions, such as whether to remain in a pluripotent state or to differentiate into specific cell types.

The impact of shear stress on stem cell differentiation is particularly noteworthy in the context of endothelial cells. Shear stress can promote the differentiation of stem cells into endothelial cells, which are essential for forming new blood vessels [71]. This process involves the upregulation of endothelial-specific genes and the downregulation of genes associated with other lineages [72]. The mechanotransduction pathways activated by shear stress led to the expression of transcription factors like Enos [73] and KLF2, which are critical for endothelial function and vascular development [74].

In addition to endothelial differentiation, shear stress has been shown to influence the differentiation of stem cells into other lineages, such as osteogenic (bone) and chondrogenic (cartilage) lineages [75]. The mechanical environment created by shear stress can enhance the expression of osteogenic markers like RUNX2 and ALP, promoting bone formation [76]. Similarly, shear force promotes chondrogenesis by regulating the differentiation of mesenchymal stem cells and maintaining chondrocyte homeostasis through mechanotransduction pathways. These pathways involve mechanoreceptors such as integrins and ion channels, which transduce mechanical signals into biochemical signals, ultimately influencing gene expression and cellular behavior [77].

The understanding of mechanotransduction in stem cells under shear stress has significant implications for tissue engineering and regenerative medicine [78]. By manipulating mechanical environments, researchers can direct stem cell differentiation and function to develop engineered tissues and organs [14]. This approach holds promise for treating various medical conditions, such as cardiovascular diseases, bone defects, and cartilage injuries.

In summary, mechanotransduction plays a pivotal role in mediating the effects of shear stress on stem cells. The conversion of mechanical signals into biochemical responses through cytoskeletal reorganization, the activation of signaling pathways, and the modulation of gene expression determines the behavior and fate of stem cells. Understanding these processes enables the development of innovative strategies for harnessing mechanical forces to direct stem cell functions for therapeutic purposes, advancing the field of regenerative medicine and tissue engineering.

## 14. Future Research Trends

The above analysis is important for predicting future research directions. The cluster analysis in Figure 8C reveals the main research clusters, including “tissue engineering”, “mesenchymal stem cells”, “expression”, “endothelial progenitor cells”, and “model”. This implies that the in-depth study of the mechanically stimulated differentiation of stem cells into specific tissues is both a current hotspot and a potential future research direction. Moreover, Figure 8B demonstrates the evolution of keywords in this field. The transition from keywords such as mechanical stress, peripheral blood, and CD34 cells to more recent terms like on-a-chip, hydrogels, and scaffolds indicates that the study of shear stress stimulation on stem cells has evolved significantly. Initially focused on understanding the effects of various mechanical stimulations on cell differentiation, the research has now advanced to incorporating sophisticated materials and technologies such as scaffolds and hydrogels. These advancements facilitate more precise and controlled studies of stem cell differentiation in response to mechanical stimulation, paving the way for innovative applications in tissue engineering and regenerative medicine. The integration of on-a-chip technologies further signifies a move towards more complex and physiologically relevant models, enhancing our ability to simulate and study the in vivo environment in a controlled in vitro setting. This progression highlights a growing interest in developing advanced biomaterials and systems that can better mimic the natural cellular environment, thereby improving the efficiency and effectiveness of stem cell-based therapies.

## 15. Limitations

Although this study can provide publication trends and hotspot predictions of shear stress stimulation on stem cells research, the limitations of this study are worth noting. First, only English-language research literature and review literature were selected for this study, while non-English databases and non-research/review literature were excluded, which may cause some research bias. Second, there was database bias in this study, as the data sources included only WoSCC and did not include Cochrane, Embase, and PubMed. In addition, high-quality literature with short publication times may have been underestimated in the bibliometric analysis due to low citation volume. Therefore, future studies should include as many types of databases as possible, analyze literature in multiple languages and types, and take certain measures to increase the weight of newly published high-quality articles to obtain the most accurate and unbiased analysis.

## 16. Conclusions

This bibliometric analysis provides a comprehensive overview of the research trends and future directions in the field of shear stress stimulation on stem cells over the past three decades. Our findings highlight a significant increase in research output, with the United States and China leading in both the quantity and quality of publications. This growth is supported by substantial funding and strong research institutions, underscoring the interdisciplinary nature of this field, particularly in areas such as materials science, engineering, cell biology, and chemistry.

High-impact journals like Biomaterials and Tissue Engineering Part A play crucial roles in disseminating significant findings, while the analysis of author collaborations reveals a robust network of researchers driving advancements. Notable contributors such as Engler AJ and Ingber DE are key figures in this research domain.

Future research is expected to focus on the mechanistic understanding of stem cell differentiation under mechanical stimulation. This includes exploring new materials and scaffold designs to better mimic the natural cellular environment. Despite the progress made, translating basic research findings into clinical applications remains a challenge. Addressing these challenges will require ongoing interdisciplinary collaboration and innovative approaches to fully harness the potential of mechanical stimulation in regenerative medicine.

This study not only maps the historical and current landscape of stem cell mechanobiology research but also identifies emerging trends and potential future directions. These insights are invaluable for researchers and policymakers in the field, guiding future investigations and fostering the development of novel therapeutic strategies.

## Figures and Tables

**Figure 1 biomedicines-12-01963-f001:**
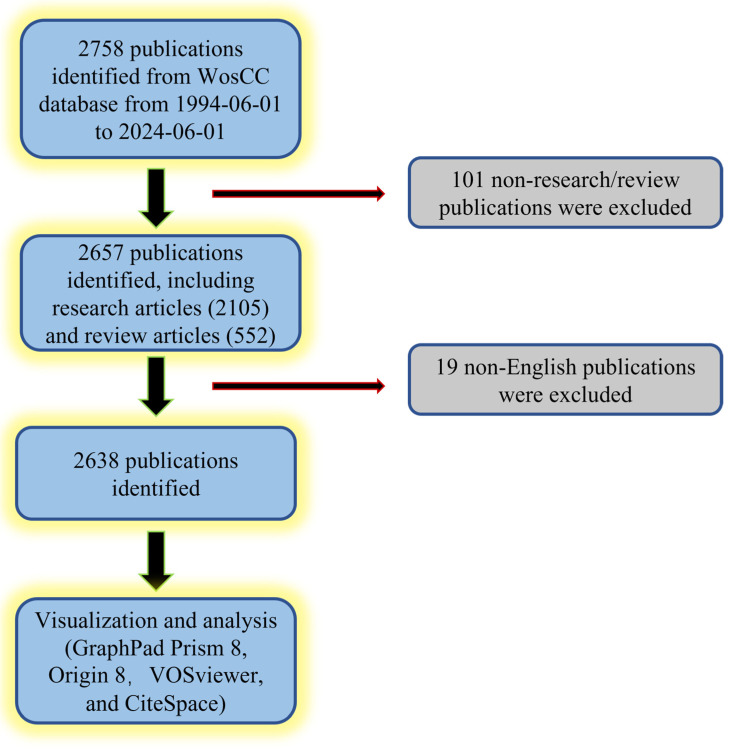
Schematic flowchart revealing the article selection process.

**Figure 2 biomedicines-12-01963-f002:**
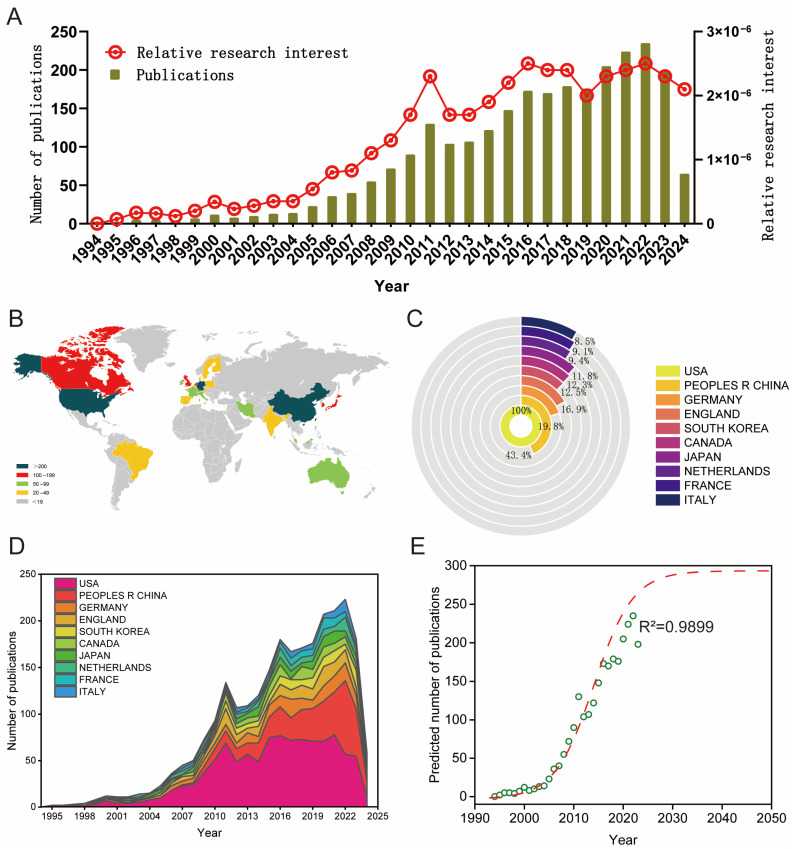
Global publishing trends and countries/regions contributing to shear stress stimulation on stem cells over the past three decades. (**A**) The annual publishing number and relative research interest (RRI) related to shear stress stimulation on stem cells. (**B**) A world map depicting the distribution of countries/regions regarding shear stress stimulation on stem cells. (**C**) The sum of publications in the top 10 countries/regions related to shear stress stimulation on stem cells. (**D**) The annual publishing number in the top 10 most productive countries/regions from 1994 to 2024. (**E**) Model fitting curves of global publishing trends in shear stress stimulation on stem cells.

**Figure 3 biomedicines-12-01963-f003:**
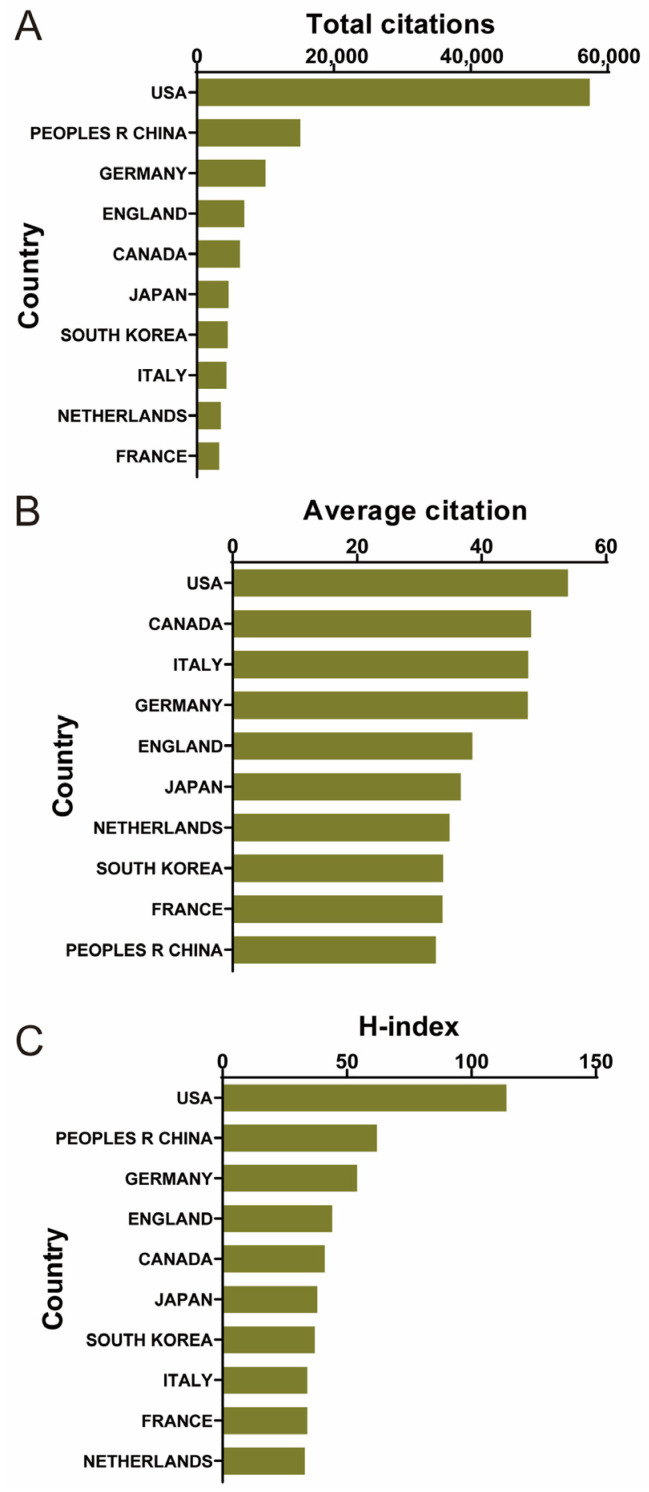
Total citations, average citations, and H-index of different countries/regions over the past three decades. (**A**) The top 10 countries and regions of total citations on shear stress stimulation on stem cells. (**B**) The top 10 countries and regions of the average citations on shear stress stimulation on stem cells. (**C**) The top 10 countries and regions of the h-index on shear stress stimulation on stem cells.

**Figure 4 biomedicines-12-01963-f004:**
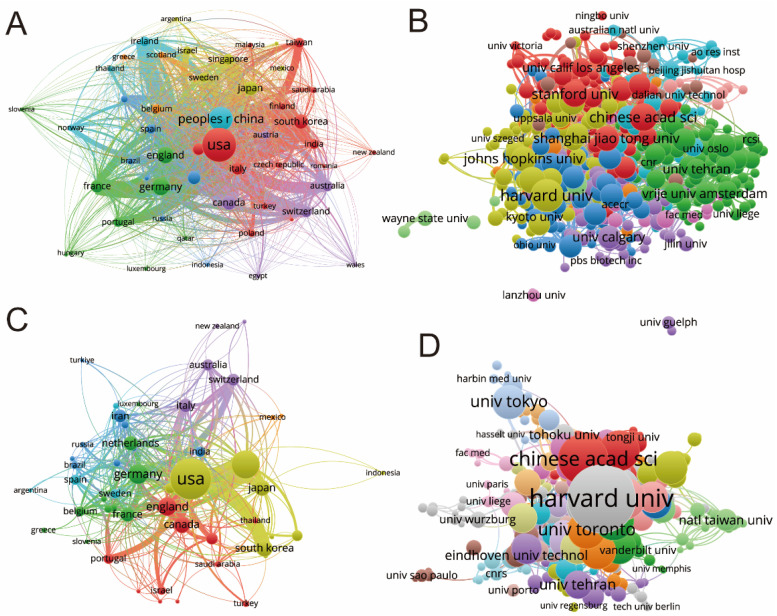
Mapping of countries/regions/institutions associated with shear stress stimulation on stem cells from 1994 to 2024. (**A**) Analysis of country/regional collaboration using VOSviewer. (**B**) Analysis of institutional collaboration using VOSviewer. (**C**) Authorship-country collaboration analysis via VOSviewer. (**D**) Authorship-institution collaboration analysis via VOSviewer. The nodes represent countries/regions, and symbolize the number of publications attributed to each node. Connecting lines denote cooperation, with thickness indicating collaboration strength where thicker lines signify closer cooperation.

**Figure 5 biomedicines-12-01963-f005:**
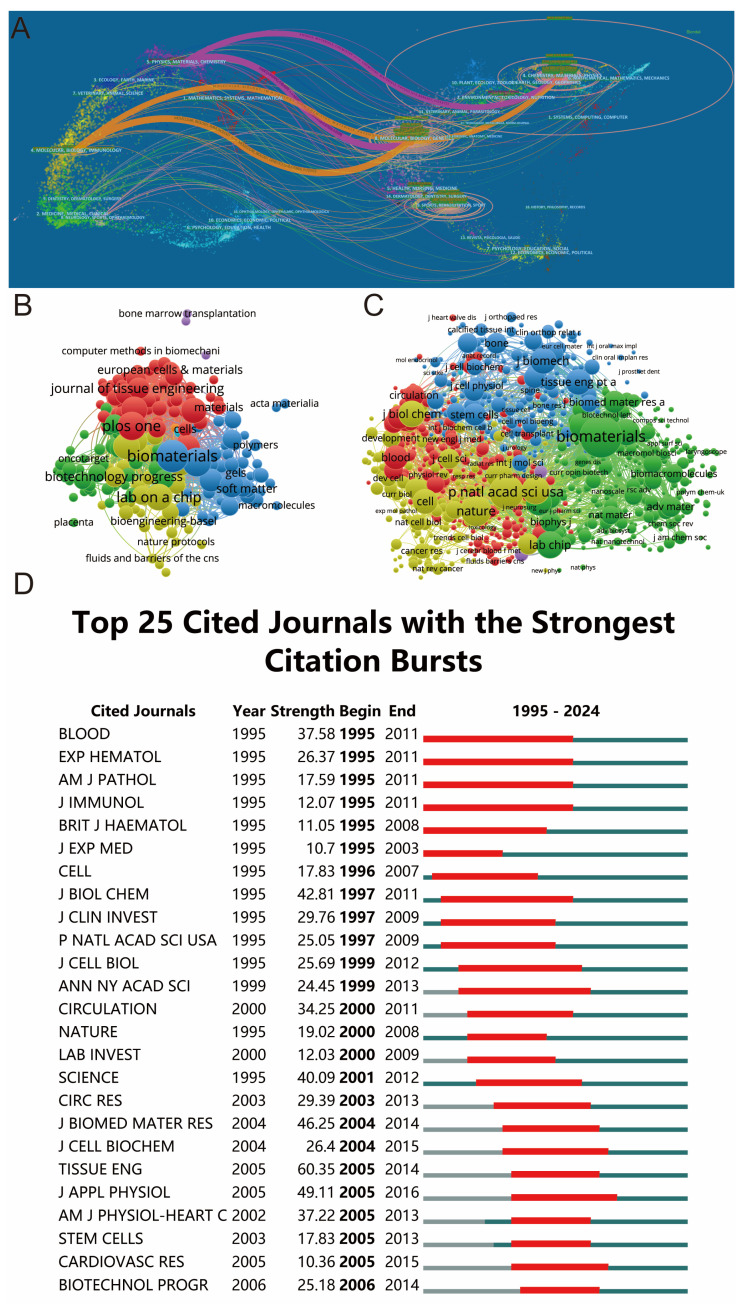
Articles published in different journals on shear stress stimulation on stem cells from 1994 to 2024. (**A**) The dual–map overlay of journals related to shear stress stimulation on stem cells. (**B**) Bibliographic of journals based on VOSviewer. (**C**) Network map of journals that were co–cited based on VOSviewer. (**D**) Top 25 cited journals with the strongest citation bursts of publications.

**Figure 6 biomedicines-12-01963-f006:**
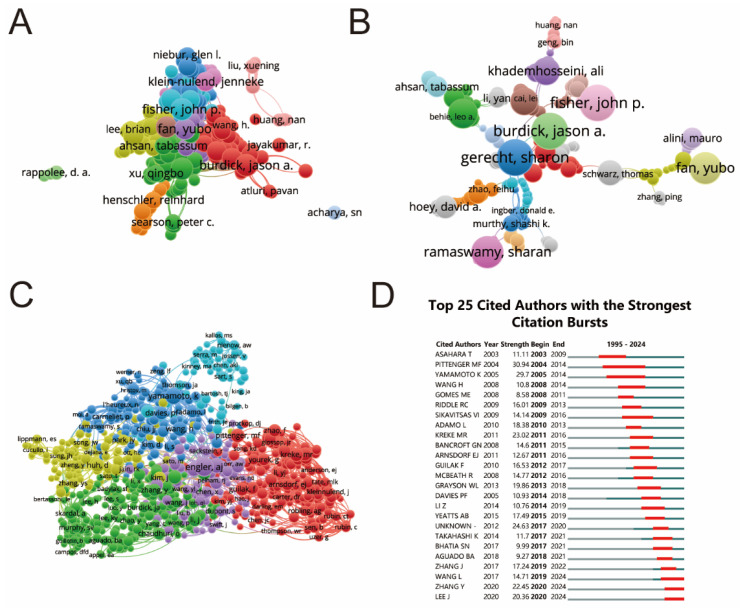
Network visualization of author collaboration analysis regarding shear stress stimulation on stem cells from 1994 to 2024. (**A**) Author collaboration analyzed by VOSviewer. (**B**) Network visualization diagram of authorship/author analysis based on VOSviewer. (**C**) Network visualization diagram of co-cited-author analysis based on VOSviewer. (**D**) Top 25 cited authors with the strongest citation bursts of publications related to shear stress stimulation on stem cells. Author collaborations are represented by nodes, with the size of each node increasing in proportion to the number of collaborations. The collaboration relationship is depicted by connecting lines between nodes.

**Figure 7 biomedicines-12-01963-f007:**
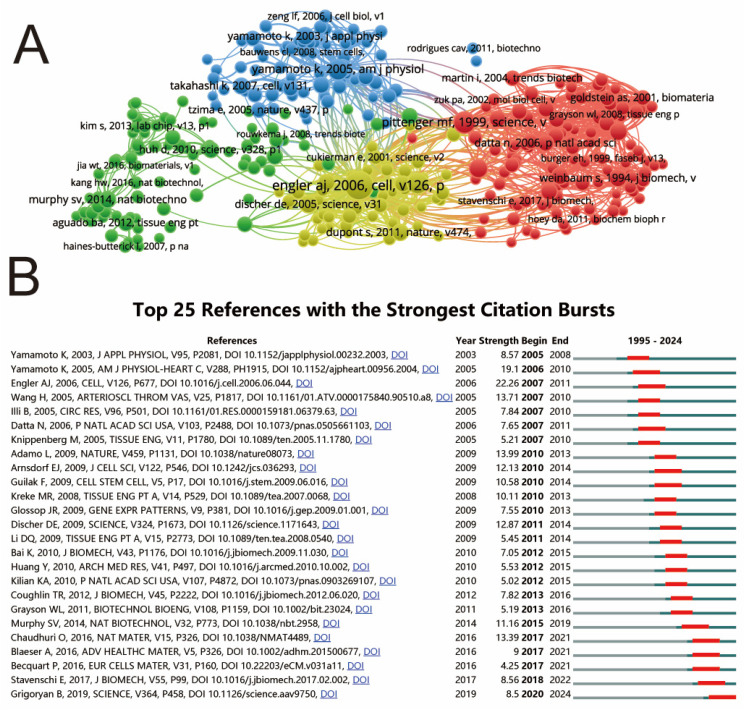
Mapping of references on shear stress stimulation on stem cells from 1994 to 2024. (**A**) Network map of references based on VOSviewer. (**B**) Top 25 references with the strongest citation bursts of publications [12,17,18,19,20,21,22,23,24,25,26,27,28,29,30,31,32,33,34,35,36,37,38,39,40].

**Figure 8 biomedicines-12-01963-f008:**
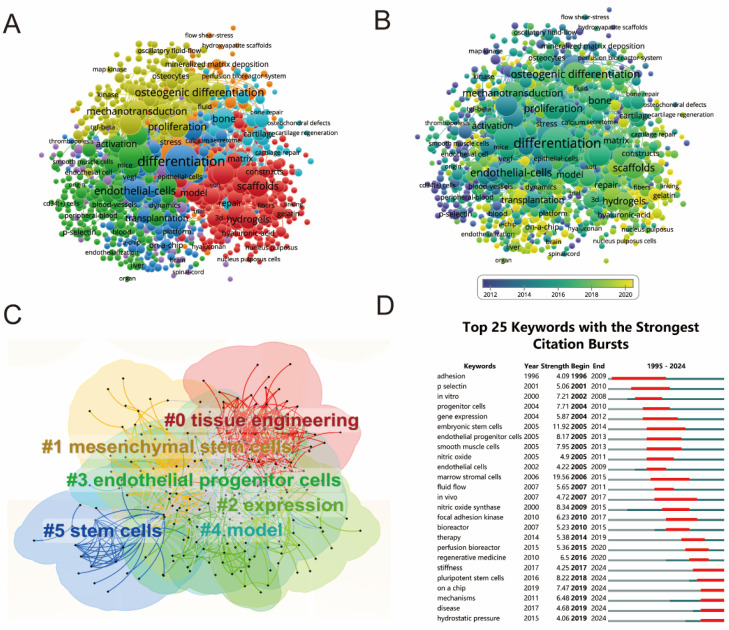
Mapping of keywords towards studies on shear stress stimulation on stem cells from 1994 to 2024. (**A**) Network visualization of keywords based on VOSviewer, and the frequency is represented by point size. (**B**) Distribution of keywords according to the mean frequency of appearance; keywords in yellow appeared later than those in blue. (**C**) Keyword clustering visualization from 1994 to 2024. (**D**) Top 25 keywords with the strongest citation bursts of publications.

**Table 1 biomedicines-12-01963-t001:** The top 10 institutions publishing literature related to shear stress stimulation on stem cells.

Rank	Institution	Article Count	Percentage (N/2638)	Country
1	University of California System	99	3.761	USA
2	Harvard University	97	3.685	USA
3	University of London	62	2.356	England
4	Harvard Medical School	60	2.28	USA
5	Brigham Women S Hospital	53	2.014	USA
6	Centre National De La Recherche Scientifique Cnrs	51	1.938	France
7	Chinese Academy of Sciences	49	1.862	China
8	University of Texas System	47	1.786	USA
9	Pennsylvania Commonwealth System of Higher Education Pcshe	44	1.672	USA
10	State University System of Florida	43	1.634	USA

**Table 2 biomedicines-12-01963-t002:** The top 10 productive journals related to shear stress stimulation on stem cells.

Rank	Journal	Article Count	Percentage (N/2638)
1	Biomaterials	71	2.698
2	Acta Biomaterialia	54	2.052
3	Plos One	51	1.938
4	Scientific Reports	45	1.71
5	Lab on A Chip	44	1.672
6	Biotechnology and Bioengineering	43	1.634
7	Tissue Engineering Part A	42	1.596
8	Journal of Biomedical Materials Research Part A	39	1.482
9	Frontiers in Bioengineering and Biotechnology	36	1.368
10	Biofabrication	35	1.33

**Table 3 biomedicines-12-01963-t003:** The top 10 well-represented research areas related to shear stress stimulation on stem cells.

Rank	Research Areas	Records	Percentage (N/2638)
1	Engineering	761	28.913
2	Materials Science	725	27.546
3	Cell Biology	516	19.605
4	Science Technology Other Topics	433	16.451
5	Biochemistry Molecular Biology	315	11.968
6	Chemistry	312	11.854
7	Biotechnology Applied Microbiology	298	11.322
8	Physics	159	6.041
9	Biophysics	154	5.851
10	Research Experimental Medicine	143	5.433

**Table 4 biomedicines-12-01963-t004:** The top 10 authors with the most publications and citations on shear stress stimulation on stem cells.

Rank	High Published Authors	Article Counts	Article Counts (N/2638)
1	Zhang Y	21	0.798
2	Alini M	16	0.608
3	Burdick JA	16	0.608
4	Fisher JP	16	0.608
5	Gerecht S	16	0.608
6	Kallos MS	16	0.608
7	Lee J	16	0.608
8	Wang Y	16	0.608
9	Chen J	15	0.57
10	Kim J	15	0.57

**Table 5 biomedicines-12-01963-t005:** The top 10 funding sources related to shear stress stimulation on stem cells.

Rank	Funds	Records	Percentage (N/2638)
1	National Institutes of Health NIH USA	596	22.644
2	United States Department of Health Human Services	596	22.644
3	National Natural Science Foundation of China NSFC	302	11.474
4	National Science Foundation Nsf	237	9.005
5	UK Research Innovation Ukri	93	3.533
6	European Union Eu	82	3.116
7	European Research Council Erc	77	2.926
8	NIH National Heart Lung Blood Institute Nhlbi	75	2.85
9	Ministry Of Education Culture Sports Science And Technology Japan Mext	74	2.812
10	Spanish Government	72	2.736

**Table 6 biomedicines-12-01963-t006:** The top five review articles with the most citations in the field of shear stress stimulation on stem cells.

Rank	Title	First Author	Journal	IF	Publication Year	Total Citations
1	Mechanobiology of YAP and TAZ in physiology and disease	Panciera, T	Nature Reviews Molecular Cell Biology	81.3	2017	791
2	The stiffness of living tissues and its implications for tissue engineering	Guimaraes, CF	Nature Reviews Materials	79.8	2020	736
3	Bioink properties before, during and after 3D bioprinting	Hölzl, K	Biofabrication	8.2	2016	693
4	Shear-thinning hydrogels for biomedical applications	Guvendiren, M	Soft Matter	2.9	2012	678
5	The Hippo Pathway: Biology and Pathophysiology	Ma, SH	Annual Review of Biochemistry	12.1	2019	668

**Table 7 biomedicines-12-01963-t007:** The top five research articles with the most citations in the field of shear stress stimulation on stem cells.

Rank	Title	First Author	Journal	IF	Publication Year	Total Citations
1	Direct 3D Printing of Shear-Thinning Hydrogels into Self-Healing Hydrogels	Highley, CB	Advanced Materials	27.4	2015	743
2	Chondrogenic differentiation of adipose-derived adult stem cells in agarose, alginate, and gelatin scaffolds	Awad, HA	Biomaterials	12.8	2004	617
3	Preferential transformation of human neuronal cells by human adenoviruses and the origin of HEK 293 cells	Shaw, G	Faseb Journal	4.4	2002	580
4	Controlling hydrogelation kinetics by peptide design for three-dimensional encapsulation and injectable delivery of cells	Haines-Butterick	Proceedings of The National Academy of Sciences of The United States of America	9.4	2007	546
5	Effect of bioink properties on printability and cell viability for 3D bioplotting of embryonic stem cells	Ouyang, LL	Biofabrication	8.2	2016	540

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
