# Peer review of "Advances in Shear Stress Stimulation of Stem Cells: A Review of the Last Three Decades"

_biomedicines, 2024, doi:10.3390/biomedicines12091963_

Round 1

Reviewer 1 Report

Comments and Suggestions for Authors

This article explores the influence of mechanical stimulation, specifically shear stress, on stem cell self-renewal and differentiation. The study systematically analyzes 2,638 research papers published between 1994 and 2024, highlighting research hotspots, publication trends, and collaboration networks. The findings underscore the growing global interest in this field, with notable contributions from the United States and China. The article also outlines promising future research directions, including a deeper exploration of mechanical stimulation mechanisms and advancements in scaffold design and regenerative medicine.

The article is well-researched, thoroughly written, and offers significant insights into the topic. The authors have effectively utilized various analytical tools to identify research trends and have clearly outlined future research directions in the field. The manuscript’s findings are both relevant and timely, contributing valuable knowledge to the ongoing discussions in this area.

I find the article to be of good quality and believe it is ready for publication without the need for any revisions.

Author Response

Dear Reviewer,

We would like to express our sincere gratitude for your thoughtful and supportive review of our manuscript. Your encouraging words regarding the depth of our research and the significance of our findings in the field of mechanical stimulation's impact on stem cell self-renewal and differentiation mean a great deal to us.

We are especially pleased that you found our analysis of the 2,638 research papers and the identification of research trends and future directions to be valuable contributions to the ongoing discourse in this area. Your feedback reassures us that our work is both relevant and timely, and it inspires us to continue our efforts in this important field.

We are grateful for your time and expertise in reviewing our manuscript and are delighted that you consider it ready for publication without further revisions.

Thank you once again for your kind words and support.

Best regards,

Yours sincerely,

Dan Xing

Arthritis Clinical and Research Center, Peking University People’s Hospital, Beijing, China; Arthritis Institute, Peking University, Beijing, China,

Reviewer 2 Report

Comments and Suggestions for Authors

The authors analyzed the literature on stem cell mechanical stimulation to predict re-search trends in the field. With this comprehensive analysis the authors enabled to identify trends in the topic about shear stress stimulation on stem cells.

Minor concerns

1.     Line 12  “Stem cells, as undifferentiated or partially differentiated cells in multicellular organisms, possess the potential for self-renewal and multi-lineage differentiation”

--- “stem cells are partially differentiated cells” could be misleading.  

2.     Line 35  “differentiates into specific types of somatic cells. [2] With their remarkable differentiation  ---  should be cells [2].  Similar errors are numerous.

3.     “P Natl Acad Sci Usa”  --- should be PNAS USA or Proc Natl Acad Sci USA

“Sci rep-uk” --- sci reports ?

4.     310 “articles about m shear stress stimulation on stem cells “ --- mechanical shear stress

5.     344, 346  “The top authors are mainly Americans”  --- should be USA or united states of America

6.     423  “mechanotransducers like YAP and TAZ, which translocate to the nucleus”  YAP and TAZ could be

7.     In the fig2 A  what does relative research interest mean compared to publication ?

8.     In the fig 3B  what is average citation ? 

9.     Table 1  “university of california system ? “  what is this ?

10.  Table 5  Nih---  NIH,  Usa--- USA,  Nsfc—NSFC

11.  143 “The number of citations and H index can reflect the quality of research”  what is H index ?  H index differs from citations ?

Author Response

Dear Reviewer,

We thank you for the time and effort that you put into reviewing the previous version of our manuscript. Your suggestions were valuable for improving the quality of our manuscript. Based on the instructions provided in your letter, we have uploaded the final revised manuscript, and all the changes are highlighted in red in the revised manuscript.

In addition, we changed the type setting of the manuscript to make it easier to read and edit and carefully proofread the manuscript to minimize the typographical and grammatical errors.

Our point-by-point responses to the comments raised by the reviewer are appended to our letter. The comments are reproduced, and our responses are provided directly after each comment in a different color (blue).

We would like to express our great appreciation to you for providing comments on our paper, and we hope our revision meets with approval. We look forward to hearing from you.

Best regards,

Yours sincerely,

Dan Xing

Arthritis Clinical and Research Center, Peking University People’s Hospital, Beijing, China; Arthritis Institute, Peking University, Beijing, China,
